# Foliar Spray of Alpha-Tocopherol Modulates Antioxidant Potential of Okra Fruit under Salt Stress

**DOI:** 10.3390/plants10071382

**Published:** 2021-07-06

**Authors:** Maria Naqve, Xiukang Wang, Muhammad Shahbaz, Sajid Fiaz, Wardah Naqvi, Mehwish Naseer, Athar Mahmood, Habib Ali

**Affiliations:** 1Department of Botany, Faculty of Sciences, University of Agriculture Faisalabad, Faisalabad 38000, Pakistan; marianaqvi26@gmail.com (M.N.); shahbazmuaf@yahoo.com (M.S.); 2College of Life Sciences, Yan’an University, Yan’an 716000, China; 3Department of Plant Breeding and Genetics, The University of Haripur, Haripur 22620, Pakistan; sfiaz@uoh.edu.pk; 4Institute of Agricultural and Resource Economics, Faculty of Social Sciences, University of Agriculture Faisalabad, Faisalabad 38000, Pakistan; wardahnaqvi5@gmail.com; 5Department of Botany, Faculty of Science and Technology, Govt. College Women University Faisalabad, Faisalabad 38000, Pakistan; mehwishnaseer@gcwuf.edu.pk; 6Department of Agronomy, Faculty of Agriculture, University of Agriculture Faisalabad, Faisalabad 38000, Pakistan; 7Department of Agricultural Engineering, Khawaja Fareed University of Engineering and Information Technology, Rahim Yar Khan, Punjab 64200, Pakistan; habib_ali1417@yahoo.com

**Keywords:** antioxidants, alpha-tocopherol, foliar spray, salinity, okra varieties

## Abstract

As an antioxidant, alpha-tocopherol (α-Toc) protects plants from salinity-induced oxidative bursts. This study was conducted twice to determine the effect of α-Toc as a foliar spray (at 0 (no spray), 100, 200, and 300 mg L^−1^) to improve the yield and biochemical constituents of fresh green capsules of okra (*Abelmoschus esculentus* L. Moench) under salt stress (0 and 100 mM). Salt stress significantly reduced K^+^ and Ca^2+^ ion concentration and yield, whereas it increased H_2_O_2_, malondialdehyde (MDA), Na^+^, glycine betaine (GB), total free proline, total phenolics, and the activities of catalase (CAT), guaiacol peroxidase (GPX), and protease in both okra varieties (Noori and Sabzpari). Foliar application of α-Toc significantly improved the yield in tested okra varieties by increasing the activity of antioxidants (CAT, GPX, SOD, and ascorbic acid), accumulation of GB, and total free proline in fruit tissues under saline and non-saline conditions. Moreover, α-Toc application as a foliar spray alleviated the adverse effects of salt stress by reducing Na^+^ concentration, MDA, and H_2_O_2_ levels and improving the uptake of K^+^ and Ca^2+^. Among the tested okra varieties, Noori performed better than Sabzpari across all physio-biochemical attributes. Of all the foliar-applied α-Toc levels, 200 mg L^−1^ and 300 mg L^−1^ were more effective in the amelioration of salinity-induced adverse effects in okra. Thus, we concluded that higher levels of α-Toc (200 mg L^−1^ and 300 mg L^−1^) combat salinity stress more effectively by boosting the antioxidant potential of okra plants.

## 1. Introduction

Okra [*Abelmoschus esculentus* (L.) Moench] is the most popular mallow crop and a common food crop in Asia. Its finger-like fruits, called capsules, are mainly consumed as a vegetable. These capsules are a rich source of vitamins, minerals and dietary fiber, and are low in calories. The mucilage’s properties have medicinal enormous value [1]. The high mucilage content sets okra apart from other vegetables and makes it suitable for various medicinal and industrial applications [2].

A 90% loss (6.5 dSm^−1^) in okra yield has been reported under high salt levels [3]. Soil salinity is one of the most prominent obstacles suppressing plant productivity. It remarkably affects the production of crops by disrupting the overall cellular metabolism of plants [4]. In Asia, it is expected that increasing levels of salinization could result in a loss of 50% of cultivated land by 2050 [5]. In developing countries like Pakistan, these losses are of considerable attention because its economy relies on agriculture. In Pakistan, 6 Mha of cultivated land is affected by salinity, which is a great threat to future food production [6].

Salinity affects plant growth and yield by reducing the photosynthesis rate, biomass, and water use efficiency [7]. The continued deposition of salts shunts osmotic stress, ionic imbalance, and physiological drought in plants [8]. The combination of these stresses directly influences fruit production in plants due to the considerable adverse effects on the composition of amino acids, proteins, and carbohydrates [9]. The production of reactive oxygen species (ROS) due to oxidation stress under high salt levels is another prominent threat as thesedamage the proteins, nucleic acids, and other biomolecules, thus limiting plant metabolism and yield [10]. This oxidative burst in the form of ROS induces the peroxidation of lipids, resulting in the production of lipid radicles and malondialdehyde (MDA), thereby damaging cellular membranes [11].

Plants have an antioxidant defense system to combat ROS. This system consists of enzymatic and non-enzymatic antioxidants found in all cellular compartments. These antioxidants detoxify cells from oxidative free radicles produced under varying saline regimes [12]. Oxidative damage generated under saline regimes can be alleviated by the exogenous application of these antioxidants [13]. Foliar spraying with α-Toc is one such approach to the improvement of plant growth under salinity stress [14]. Tocopherols are members of the vitamin E family and consist of alpha, beta, gamma, and delta forms [14]. α-Toc is more active than all other categories of vitamin E, as it protects photosystem II and lipid membranes in chloroplasts from salinity-induced damages [13,14].

Tocopherols are non-enzymatic antioxidants which protect plants by quenching ROS and guard cellular membranes against lipid peroxidation [15]. Among these tocopherols, α-Toc is the most active antioxidant as it shields photosystems from photo-inhibition and protects membrane lipids in chloroplasts under salinity stress [9]. As chloroplasts are sensitive to salinity stress [16], to combat salinity-induced ROS, α-Toc works in coordination with other antioxidants, including catalase (CAT), superoxide dismutase (SOD), and guaiacol peroxidase (GPX) [17]. SOD is the first line of defense under stress conditions as it converts singlet oxygen species to H_2_O_2_, and this H_2_O_2_ is converted to H_2_O by CAT and GPX [18].

Few studies have assessed the impact of foliar spraying with α-Toc in boosting the antioxidant potential of okra fruit under salt stress. Therefore, this study aimed to examine the modulations in antioxidant defense mechanism in response to α-Toc foliar spray on the yield and the related attributes of okra under salt stress conditions.

## 2. Results

The data revealed that catalase (CAT) activity was significantly increased under salinity stress. Neither of the tested varieties of okra differ significantly in terms of CAT activity. Foliar spraying (300 mg L^−1^) with α-Toc enhanced CAT activity in okra capsules under saline and non-saline conditions (Table 1; Figure 1A).

Both okra varieties demonstrated non-significant performance in term of peroxidase (POD) activity. Neither the application of salt stress nor α-Toc affected the POD activity of the okra capsules. The data showed that root medium salinity had no effect on the activity of protease. However, a significant interaction was recorded between salinity and the α-Toc spray, where 300 mg L^−1^ of the spray proved effective to increase the activity of POD under saline conditions (Table 1; Figure 1B).

The experimental data showed that root medium 100 mM saline stress significantly enhanced the activity of guaiacol peroxidase (GPX) in the fruit tissues of both okra varieties. However, in the salinized Noori plants, GPX activity was slightly higher than Sabzpari. In addition, α-Toc spray markedly improved the activity of GPX in both varieties. Inclusively, the 200 mg L^−1^ application of α-Toc proved to be better at enhancing the activity of GPX (Table 1; Figure 2A).

A slight increase in the activity of superoxide dismutase (SOD) in the fruits of both varieties of okra plants was noticed due to salinization. Interestingly, levels of superoxide dismutase (SOD) were higher in Noori than Sabzpari capsules under both stressed and non-stressed conditions. In the current study, foliar-applied α-Toc failed to significantly affect the activity of SOD (Table 1; Figure 2B).

In this study, a significant decrease was recorded in the activity of protease under salinity stress. It was observed from the data that Noori performed better than Sabzpari with respect to this attribute. A significant interaction was also recorded among the tested varieties. The application of 200 mg L^−1^ spray of α-Toc under saline conditions significantly enhanced the protease content of Noori okra in comparison with other treatments (Table 1; Figure 3A).

A significant accumulation of ascorbic acid was recorded under salinity stress in the Noori variety, while it remained the same in Sabzpari under saline and control conditions. The application of α-Toc significantly enhanced the ascorbic acid content in the Noori variety under saline conditions. Among all treatments, α-Toc concentrations of 200 mg L^−1^ and 300 mg L^−1^ were more effective with respect to this attribute in the case of the Noori variety (Table 1; Figure 3B).

The data revealed that the total phenolics content significantly increased under salt stress. The tested varieties differed significantly in terms of total phenolics content. Noori produced more Sabzpari. Higher levels of applied α-Toc (200 mg L^−1^ and 300 mg L^−1^) spray exhibited clearly higher total phenolics content under saline regimes (Table 1; Figure 4A).

Both okra varieties have shown non-significant differences in terms of fruit glycine betaine (GB) and free proline contents. Root medium salinity stress non-significantly enhanced GB content (Table 2; Figure 4B,C) while markedly enhancing free proline in both okra varieties. Foliage supplementation with α-Toc did not markedly affect the proline content of the fruit, whereas a significant interaction was observed between α-Toc spray and salt stress with respect to the accumulation of proline content. Higher levels of α-Toc spray produced significantly higher proline content under 100 mM salt stress conditions as compared to other combinations. In addition, foliar spraying with α-Toc at 200 and 300 mg L^−1^ significantly increased GB content in okra fruits of both varieties under controlled and stressed environments (Table 2; Figure 4B,C).

Interestingly, neither the foliar application of α-Toc nor the application of salt (NaCl) showed a clear impact on hydrogen peroxide (H_2_O_2_) content. However, both okra varieties differed significantly in terms of hydrogen peroxide concentrations; Sabzpari accumulated more H_2_O_2_ than Noori under salt stress and non-stressed conditions (Table 2; Figure 5A).

Malondialdehyde (MDA) content was markedly enhanced under salt stress conditions in the Sabzpari variety. Foliar spraying with α-Toc significantly reduced MDA contents under salinity treatment conditions. The experimental data of this study showed that higher levels of α-Toc (200 mg L^−1^ and 300 mg L^−1^) were more effective at reducing the MDA content of okra (Table 2; Figure 5B).

Root medium applied salt stress was found to significantly enhance the fruit Na^+^ content of both okra varieties. The response of the two okra varieties was non-significant in terms of fruit Na^+^ content. Foliar spraying with α-Toc failed to have a significant impact on this ionic attribute, but the interactive effect of both varieties and α-Toc was significant; it was found that 200 and 300 mg L^−1^ of foliar spray decreased Na^+^ content in both varieties under saline and non-saline regimes. Noori performed better than Sabzpari in lowering the Na^+^ ion content under salinity stress (Table 2; Figure 6A).

In this study, a clear reduction in fruit K^+^ content was determined in both okra varieties under saline conditions. Although both varieties have shown significant differences in terms of fruit K^+^ content, the Noori variety proved to be better at accumulating more K^+^ than Sabzpari under both salinity treatments. However, a significant interactive effect between α-Toc and salinity stress was noted in this study. Higher levels of α-Toc spray enhanced fruit K^+^ content under salinity stress (100 mM NaCl) conditions in comparison with other combinations (Table 2; Figure 6B).

Repeated experimental data showed that soil salinization remarkably decreased fruit Ca^2+^ content in both okra varieties. Noori had higher fruit Ca^2+^content than Sabzpari under the salinity-free regime. The interactive effect of both varieties towards salinity was also significant. Fruit Ca^2+^ content increased remarkably after foliar application of α-Toc. The response of α-Toc under salinity stress was also significant. Inclusively, 300 mg L^−1^ increased the fruit Ca^2+^ content in tested varieties under saline and no-saline environments (Table 2; Figure 6C).

Salt stress significantly reduced the yield of both okra varieties in terms of the number of capsules plant^−1^. Both varieties differed markedly though; yield reduction was more pronounced in Sabzpari than Noori. However, foliar application of α-Toc significantly enhanced the number of capsules in both okra varieties of stressed and non-stressed plants. In addition to this, enhancement in yield was more pronounced at 200 mg L^−1^ and 300 mg L^−1^ levels of α-Toc application (Table 2; Figure 7).

## 3. Discussion

Salinity is considered an abiotic stress due to its potential to harm various physiological and biochemical processes in plants [19,20]. It influences plant performance, limiting its production and causing water shortages to the plants. Salinity also negatively affects chlorophyll, the photosynthetic apparatus, and chloroplast ultra-structure, and may cause cell death [21,22]. Salinity-induced ROS triggers phytotoxic reactions and adversely affects the cellular processes of plants, causing tissue damage due to the oxidization of macro-molecules such as proteins and lipids. The induction of antioxidants is the prominent activity plants use to combat salinity-induced ROS stress [16]. Antioxidant enzymes are recognized as effective mitigators of the adverse effects of salt stress on cells and tissues [9]. The results of this study suggest that the activities of antioxidants, including CAT, GPX, and protease, were significantly increased due to salt stress in both varieties of okra plants. A slight but non-significant increase was observed in POD activity due to salt stress. The GPX is considered to be a defensive agent against oxidation induced by H_2_O_2_ [23], while POD is as efficient as GPX at scavenging H_2_O_2_ under salt stress conditions [24]. Previous research has shown that catalase plays an important role in ROS detoxification, converting 26 million H_2_O_2_ to H_2_O in one minute [25].

In the current study, the activities of these enzymes viz. CAT, POD, GPX, and protease were enhanced by the foliar fertigation of α-Toc in fruit tissues of the tested okra varieties. To combat the overproduction of ROS, α-Toc aids in the coordination of CAT, POD and GPX [26]. SOD activity was also increased in tested varieties of okra plants grown in saline medium, but α-Toc spray did not significantly increase its activity. The enzyme SOD is considered to be the first line of defense against ROS. In a previous study, the activities of SOD, POD, and CAT were increased in two mung bean varieties under stress conditions [27]. Similarly, a study of *Brassica napus* showed a remarkable enhancement in the activity of SOD and POD enzymes, while a decline was noticed in the activity of CAT under stress conditions [28]. It is concluded from the findings of the present study that salt-stress-induced ROS accumulation is scavenged by α-Toc antioxidants as they prevent ROS production by chelating metal ions. Moreover, α-Toc fertigation could compensate for salinity-induced damages by upregulating other enzymatic antioxidants and quenching ROS via its antioxidant properties. Researchers have concluded that the stress-induced ROS accumulation is dependent on antioxidants, including α-Toc [29].

Foliar spraying with α-Toc improved ascorbic acid and total phenolic content in the tested okra plants under salt stress. It is suggested that ascorbic acid protects the plants against salinity effects by acting as a free radical reductant and antioxidant, and salt tolerance may have also been manifested by the upregulation of ascorbic acid, as it induces Toc biosynthesis and total phenols provide membrane stability against salinity-induced ROS. There is a positive correlation between spraying with α-Toc and the acceleration of the content of total phenolics [30].

Glycine betaine (GB) has a significant role in stabilizing the structure and activity of protein complexes and enzymes and also maintains the membrane against devastating impacts caused by salinity [31]. Proline is effective in maintaining the potential of the osmotic process in the cytoplasm and in maintaining proteins and ribosomes against the harmful effects of Na^+^ ions. In the current study, salt stress did not significantly affect the proline and GB content in fruit tissues of okra varieties. Contrary to this, previous research has shown a remarkable increase in the accumulation of GB and proline under stress [31]. Foliar spraying with α-Toc significantly enhanced the proline and GB content, indicating its role in shielding the plants from salinity effects by adjusting the osmotic balance.

The ROS can induce the peroxidation of lipids (in MDA), producing aldehydes, of which MDA is a major type. MDA consequently has a role as an indicator of membrane damage induced by ROS. Our findings showed that neither the foliar application of α-Toc nor salt (NaCl) significantly affected H_2_O_2_ content, whereas MDA contents were increased under salt stress conditions in both okra varieties. In the current study, foliar spraying with α-Toc significantly reduced MDA content under salinity stress in the tested okra capsules. Contrary to the present study, lower levels of H_2_O_2_ had been observed in *Vicia faba* plants treated with α-tocopherol during salinity stress [14]. This could be due to the variation in genetic traits among different plant species. However, α-Toc helped stabilize lipid membranes by scavenging ROS [32].

In this study, higher concentrations of Na^+^ ions had been observed in the fruit tissues of okra plants under saline regimes. Enhanced Na^+^ concentration disturbed photosynthesis, plant metabolism, enzymatic activities, and ultimately reduced crop yield [33,34]. In contrast, reduced uptake of K^+^ and Ca^2+^ had been observed in the fruit tissues of the tested okra plants under salt stress conditions. The reduced uptake of Ca^2+^ and K^+^ is correlated with the enhanced concentration of Na^+^. α-Toc spray proved to be successful in minimizing the toxic concentrations of Na^+^ and maximizing the concentration of K^+^ and Ca^2+^ ion contents in fruit tissues of okra plants, suggesting its pivotal role in osmo-tolerance by lowering Na^+^ levels and enhancing the uptake of K^+^ and Ca^2+^ ions in plants under salinity conditions.

Plants are the target of abiotic stresses, thus suffering severe yield losses [35]. In the current investigation, reduced numbers of capsules were observed in okra under salinity stress. A reduction in the yield of crop plants under salinity stress is directly linked with inhibited uptake of essential nutrients [36]. However, α-Toc supplementation significantly improved yield under saline and non-saline conditions. Increased yield due to α-Toc spray is linked with the improved uptake of beneficial ions, ionic homeostasis, chloroplast stability, and reduced oxidative damage.

## 4. Material and Methods

Two-year pot experiments were carried out to examine the possible role of α-Toc in modulating the antioxidant potential of okra capsules under salinity stress. Experiments were conducted in plastic pots (diameter 26 cm, depth 29 cm) each containing 10 kg of well washed river sand. The number of plots was 64 for each experiment. The crop duration was six months from sowing to harvesting. Each experiment was Seeds of two okra varieties (Sabzpari and Noori) were collected from the Ayub Agricultural Research Institute, Faisalabad, Pakistan. Experiment was laid out under completely randomized design (CRD) with four replications. Hoagland’s nutrient solution (1 L/pot) was applied weekly after sowing. Five plants were maintained in each pot. Twenty-four days old plants were treated with two salt levels (0 and 100 mM NaCl) in Hoagland’s nutrient solution. Concentration of NaCl was maintained in aliquot parts of 50 mM to prevent salt shock. In fact, in order to avoid osmotic shocks to plants, the salinity level was maintained in two phases. In the first phase (at start of salinity) 50 mM NaCl level was maintained, and after two days at this level, the required level of 100 mM was maintained. Foliar spray of each concentration of α-Toc (0, 100, 200 and 300 mg/L) was applied to 36 days old plants. The 50 mL solution of each of α-Toc levels was foliar sprayed to fully saturate the plants. Tween 20 at 0.1% was used to enhance the absorbance of solution as surfactant. All plants were allowed to grow until the complete formation of capsules. Fresh green capsules of okra plants were collected from each plant, weighed, and used for chemical analyses as described below.

### 4.1. Estimation of Enzymatic Antioxidants

Fresh green capsule material (0.5 g) was homogenized with 10 mL (50 mM) potassium phosphate buffer with pH 7.0, after centrifugation supernatant was used for the determination of enzymatic antioxidants; activity of CAT and POD was determined by following the protocol of Chance and Maehly [37]. The procedure of Giannopolitis and Ries [36] was followed for recording the activities of SOD.

Protocol ascribed by Carlberg and Mannervik [38] was followed for the determination of GPX activities. Protease activities were recorded by following Drapeau et al. [39].

### 4.2. Estimation of Non-Enzymatic Antioxidants

For the determination of total phenolics content from the fresh green tissue of capsules, the protocol of Ainsworth and Gillespie [40] was followed. Ascorbic acid content from the capsules tissues of okra was determined by following the protocol established Mukherjee and Choudhuri [41]. Similarly, using the protocol established by Grieve and Grattan [42], glycinebetaine content was estimated from fresh fruit tissue. 

The protocol ascribed by Bates et al. [43] was followed to estimate the free proline from the fruit material.

### 4.3. Determination of ROS

The procedure ascribed by Alexieva et al. [44] was adopted for the determination of H_2_O_2_ level, and by following procedure of Heath and Packer [45] with slight modifications [46] malondialdehyde (MDA) content was estimated.

### 4.4. Nutrients Analysis for Na^+^, K^+^ and Ca^2+^

The protocol proposed by Wolf [47] was followed to determine mineral elements by acid digestion.

### 4.5. Yield

The plucked fresh capsules were collected, and yield (number of capsules/plants) was recorded.

### 4.6. Statistical Analysis

By following Snedecor and Cochran [48], analysis of variance (ANOVA) of data for all the parameters with four replicates was calculated under three factor factorial and design of experiment was completely randomized (CRD). The least significance difference (LSD) values at 5% probability were worked out and are presented in each figure.

## 5. Conclusions

The foliar fertigation of α-Toc was effective in compensating harmful salinity-induced effects in okra by enhancing antioxidant activities (CAT, POD, SOD, GPX, protease, ascorbic acid, total phenolics) and organic osmolytes (GB, free proline), as well as by improving ionic homeostasis and yield, possibly by quenching salinity-induced ROS and protecting chloroplasts with its antioxidant potential. Among the tested okra varieties, Noori showed better tolerance against salinity. α-Toc levels of 200 and 300 mg L^−1^ were more effective. Thus, this study suggests the Noori variety can be grown in salt-affected soils with α-Toc foliar spray (300 mg L^−1^) to increase okra production. The brutal effects of salinity can be mitigated in okra as well as other crops by foliar spraying with α-Toc.

## Figures and Tables

**Figure 1 plants-10-01382-f001:**
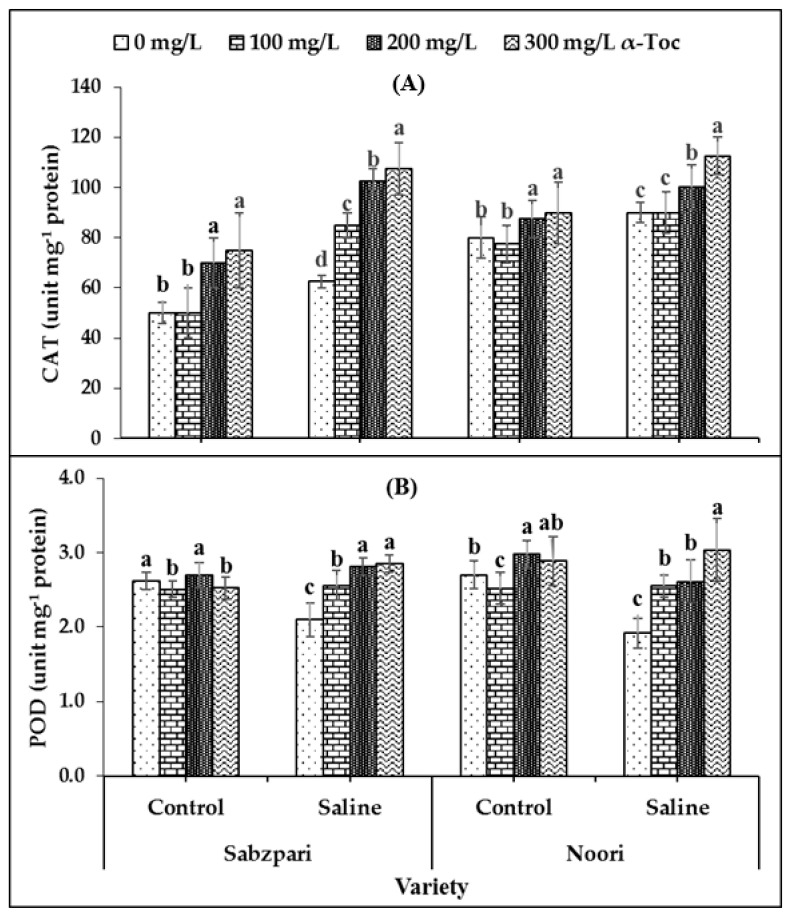
Effect of different levels of α-Toc under saline and non-saline conditions on (**A**) activities of catalase (CAT) and (**B**) peroxidase (POD) of okra varieties sprayed with different levels of α-Toc under saline and non-saline conditions. Values represent means ± S.D. Significant differences among row spacing were measured by the least significant difference (LSD) at *p* > 0.05 and indicated by different letters. S.D. stands for standard deviation.

**Figure 2 plants-10-01382-f002:**
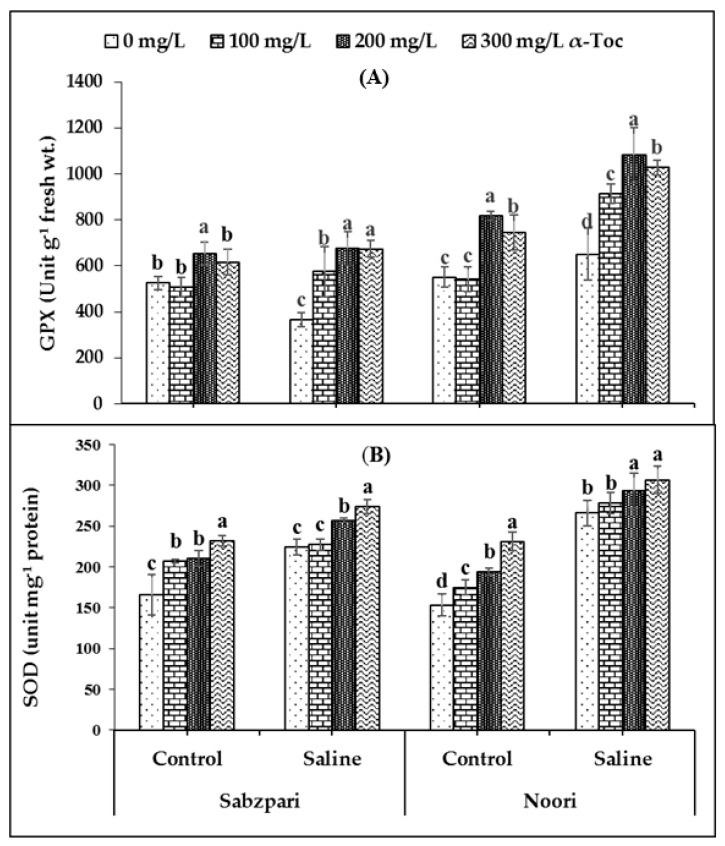
Effect of different levels of α-Toc under saline and non-saline conditions on (**A**) activities of guaiacol peroxidase (GPX) and (**B**) superoxide dismutase (SOD) of okra varieties. Values represent means ± S.D. Significant differences among row spacing were measured by the least significant difference (LSD) at p > 0.05 and indicated by different letters. S.D. stands for standard deviation.

**Figure 3 plants-10-01382-f003:**
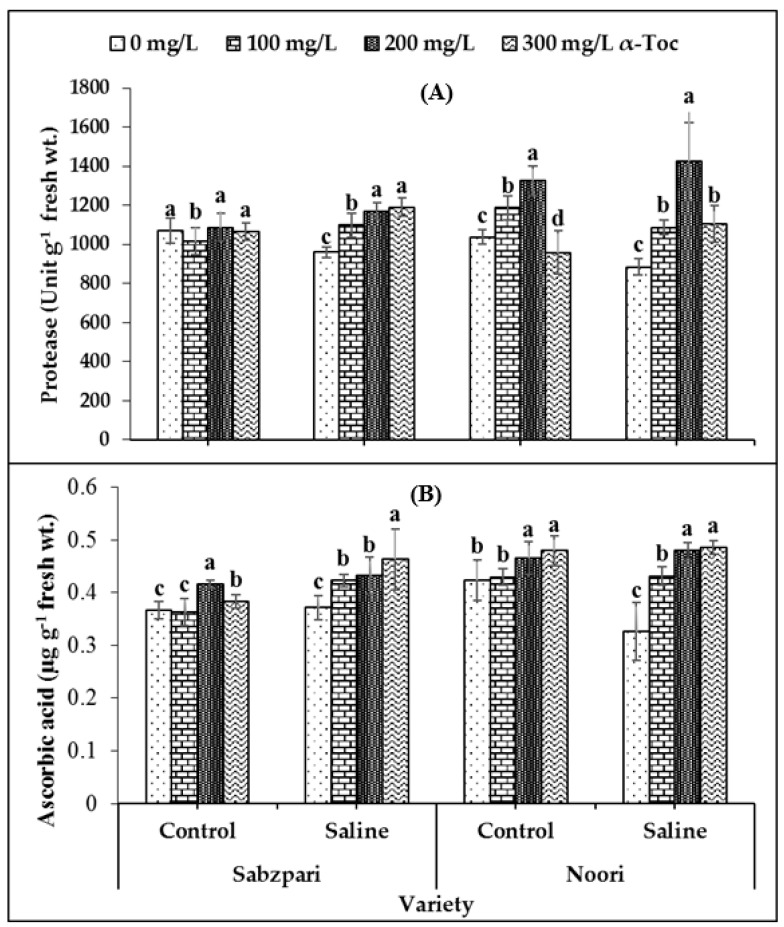
Effect of different levels of α-Toc under saline and non-saline conditions on (**A**) activities of protease and (**B**) ascorbic acid of okra varieties. Values represent means ± S.D. Significant differences among row spacing were measured by the least significant difference (LSD) at *p* > 0.05 and indicated by different letters. S.D. stands for standard deviation.

**Figure 4 plants-10-01382-f004:**
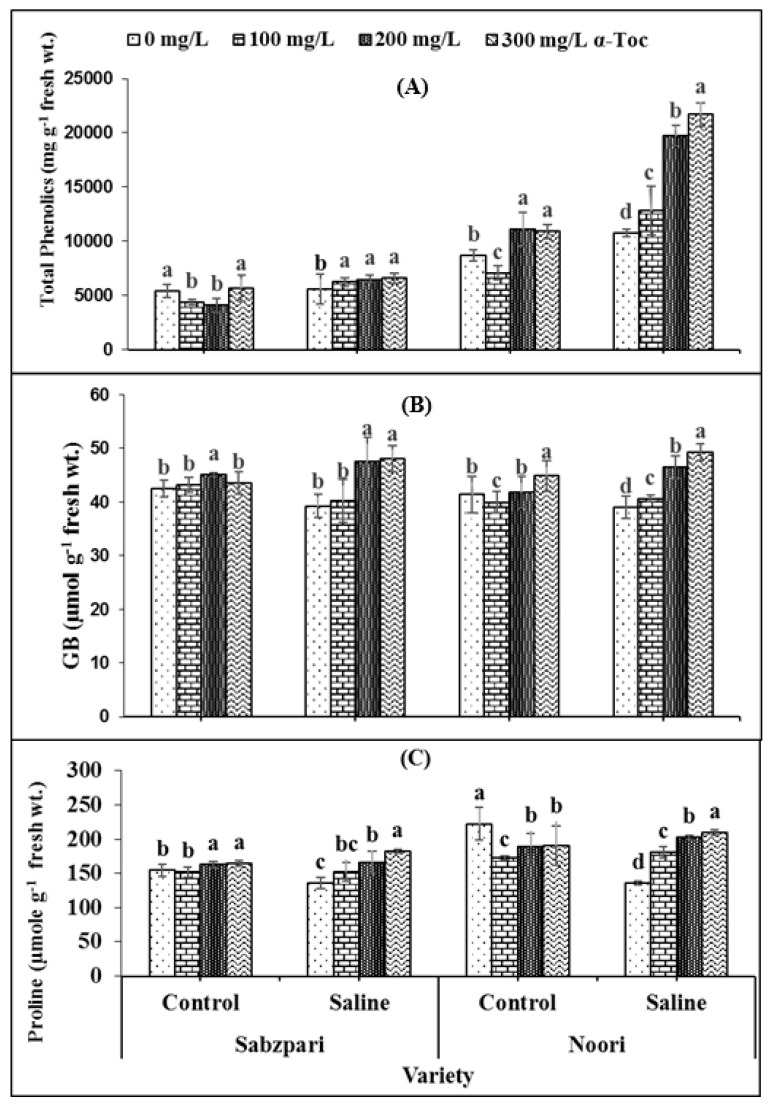
Effect of different levels of α-Toc under saline and non-saline conditions on (**A**) total phenolics (**B**) glycinebetaine (GB), and (**C**) proline content of okra varieties. Values represent means ± S.D. Significant differences among row spacing were measured by the least significant difference (LSD) at *p* > 0.05 and indicated by different letters. S.D. stands for standard deviation.

**Figure 5 plants-10-01382-f005:**
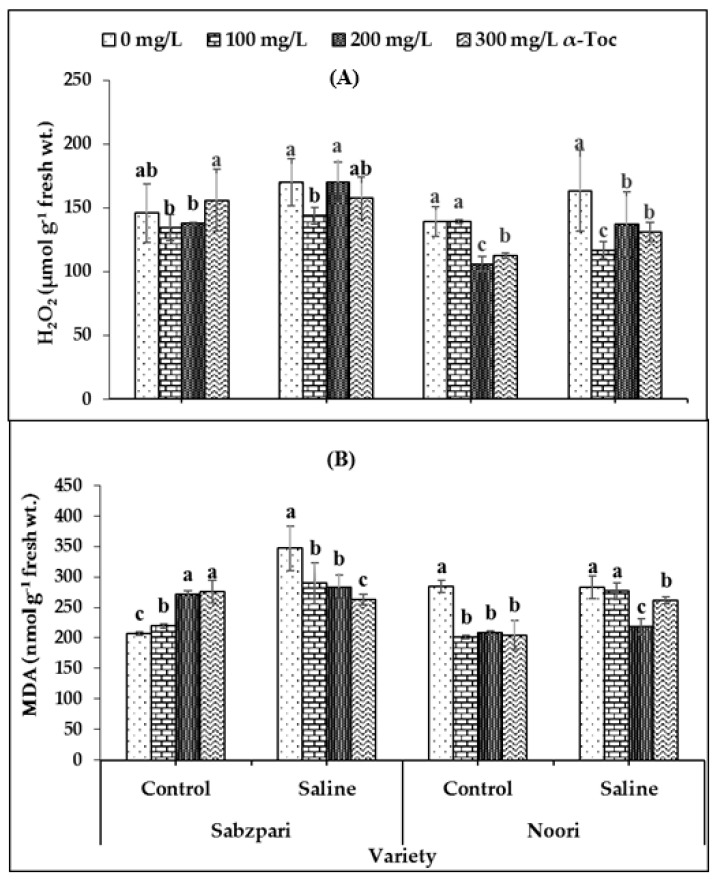
Effect of different levels of α-Toc under saline and non-saline conditions on (**A**) hydrogen peroxide (H_2_O_2_) and (**B**) malondialdehyde (MDA) content of okra varieties. Values represent means ± S.D. Significant differences among row spacing were measured by the least significant difference (LSD) at *p* > 0.05 and indicated by different letters. S.D. stands for standard deviation.

**Figure 6 plants-10-01382-f006:**
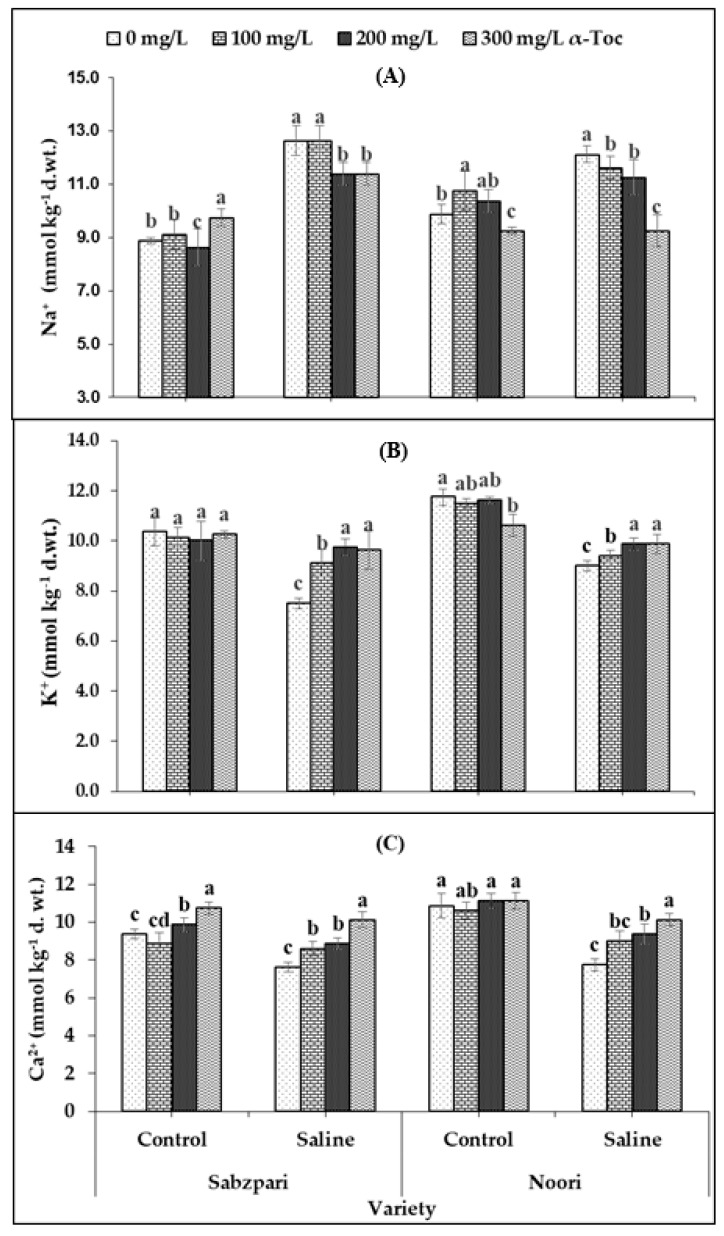
Effect of different levels of α-Toc under saline and non-saline conditions on (**A**) Na^+^ (**B**) K^+^ and (**C**) Ca^2+^ ion content of okra varieties. Values represent means ± S.D. Significant differences among row spacing were measured by the least significant difference (LSD) at *p* > 0.05 and indicated by different letters. S.D. stands for standard deviation.

**Figure 7 plants-10-01382-f007:**
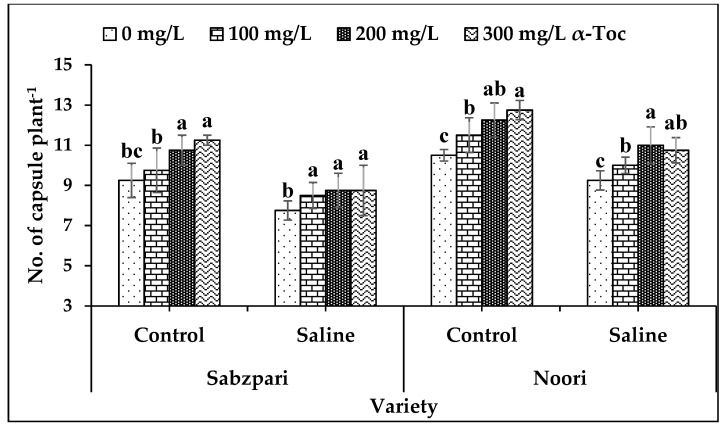
Effect of different levels of α-Toc under saline and non-saline conditions on number of capsule Plant^-1^ of okra varieties. Values represent means ± S.D. Significant differences among row spacing were measured by the least significant difference (LSD) at *p* > 0.05 and indicated by different letters. S.D. stands for standard deviation.

**Table 1 plants-10-01382-t001:** Analysis of variance (mean squares) for enzymatic and non-enzymatic antioxidants traits of okra treated with α-Toc as foliar spray under saline and non-saline conditions.

Source	df	CAT	POD	SOD	GPX	Protease	Phenolics	Ascorbic Acid
V	1	264.06 ns	0.266 ns	119.52 **	690,058.34 ***	28,532.84 ***	1.22 ***	1388.89 ns
S	1	15,314.06 ***	0.548 ns	54.83 ns	306,132.59 ***	8883.06 ***	293,265,625 ***	41,769.14 ***
α-toc	3	2143.22 **	0.515 ns	16.29 ns	279,930.76 ***	607.43 ns	1.32 ***	4013.78 *
V × S	1	1501.56 ns	0.523 ns	47.23 ns	290,599.16 ***	70.84 ns	3.94 ***	28,110.11 ***
V × α-toc	3	489.06 ns	0.065 ns	21.94 ns	18,401.16 ns	48,183.32 ***	6,912,135.4 ns	3462.61 *
S × α-toc	3	205.72 ns	0.997 **	6.42 ns	4860.57 ns	18,282.91 ***	32,911,354 **	16,191.37 ***
V × S × α-toc	3	801.56 ns	0.109 ns	24.20 ns	57,395.68 *	7053.95 ***	9,296,354.2 ns	13,829.37 ***
Error	48	403.64	0.184	15.23	18,769.04	404.83	6,673,880.2	1109.76

*, ** and *** = significant at 0.05, 0.01 and 0.001 levels respectively, ns = non-significant, V: Varieties, S: Salinity, α-Toc: Alpha-tocopherol, CAT: Catalase, POD: Peroxidase, SOD: Superoxide dismutase, GPX: Guaiacol peroxidase.

**Table 2 plants-10-01382-t002:** Analysis of variance (mean squares) for osmolytes, ROS, ionic traits and yield of okra treated with α-Toc as foliar spray under saline and non-saline conditions.

Source	df	Proline	GB	H_2_O_2_	MDA	Na^+^	K^+^	Ca^2+^	cap/pl
V	1	992.25 ns	0.33 ns	7284.92 *	228.39 ns	0.19 ns	11.81 ***	8.6 **	43.89 ***
S	1	10,905.3 **	316.19 ns	3519.39 ns	25,656.80 ***	53.47 ***	36.75 ***	30.9 ***	43.88 ***
α-toc	3	1983.47 ns	1097.5 **	1344.55 ns	5587.57 **	1.11 ns	1.19 ns	7.9 ***	9.515 **
V × S	1	2102.87 ns	5.3 ns	63.87 ns	8050.72 **	18.59 ***	1.72 ns	3.7 *	0.390 ns
V × α-toc	3	2973.15 *	313.3 *	840.67 ns	12,923.24 ***	6.34 ***	0.84 ns	0.6 ns	0.182 ns
S × α-toc	3	3381.21 *	548.0 **	1129.41 ns	2793.45 ns	0.79 ns	3.51 **	2.2 *	0.682 ns
V × S × α-toc	3	9211.3 ***	9.3 ns	404.06 ns	3888.35 *	0.96 ns	0.60 ns	0.2 ns	0.182 ns
Error	48	1019.95	86.6	1014.53	1044.26	0.98	0.70	0.7	2.234

*, ** and *** = significant at 0.05, 0.01 and 0.001 levels respectively, ns = non-significant, V: Varie Table 2. O_2_, Hydrogen peroxide, MDA: Malondialdehyde, cap/pl: Capsules per plant.

## Data Availability

Not applicable.

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
