# Peer review of "Foliar Spray of Alpha-Tocopherol Modulates Antioxidant Potential of Okra Fruit under Salt Stress"

_plants, 2021, doi:10.3390/plants10071382_

Round 1
Reviewer 1 Report
The manuscript of paper by Naqve M. et al. “Foliar spray of alpha-tocopherol modulates antioxidant potential of okra fruit under salt stress” is devoted to the study of fertigation of vitamin E on the Abelmoschus esculentus plant to cope with salinity effects, mainly oxidative stress effects. Authors provided set of data obtained on antioxidant enzymes activities, antioxidant and osmolyte content as well as ion homeostasis upon α-tocopherol treatment under saline stress. Unfortunately, data are presented in unclear form and rather poorly discussed that in many cases it is very difficult to agree with author’s interpretation (see below). The manuscript contains lots of misformattings and design of figures, in spite of rather good English spelling.
1) There are several inaccuracies in the abstract. It is written ‘Current study was conducted twice to determine…’ (line 19). What does it mean? In ‘Material and Methods’ section it is said that there were 4 replicates, neither two. Different parameters were mentioned to alter ‘in both okra varieties’ (line 24). It is better to specify, what varieties? Abstract contains several missing spaces (lines 26, 31, 32).
2) Should not the ‘Material and Methods’ section be placed in the end part of the article before the ‘Conclusion’ section? Also there is a mistake in the word ‘methods’.
3) Authors use several variants of ‘α-tocopherol’ mentions throughout the text (alpha tocopherol, alpha-tocopherol and α-Toc). It is better to unify the variant and use the abbreviation since the first mention, and not from the second page of the text.
4) Line 44, does (6.5 dsm-1) mean (6.5 dSm-1)? If so, it has to be corrected.
5) There are several incorrect uses of uppercase letters in the text (lines 65, 263).
6) Information on lines 74-77 of ‘Introduction’ is somewhat similar to the information on lines 69-73 and has to be rearranged.
7) It is unclear, how 100 mM concentration of NaCl) was maintained in aliquot parts of 50 mM (lines 94-96).
8) What is ‘Tween 20 @ 0.1%was’? There is missing space between ‘%’ and ‘was’ as well.
9) Lines 104-105, plant material / buffer ratio and buffer pH are not specified.
10) Authors measured content of a number of antioxidants, but not α-tocopherol itself. Also glycine betaine can hardly be attributed to antioxidants.
11) Lines 126-128: Authors counted the number of capsules per plant in terms of yield assessment. Why not to measure capsules weight?
12) The results of the study are not clearly presented. There are 2 tables and 2 figures with excessive information given in absolutely reader-unfriendly way. It is better to split figures into several ones and enlarge their size. The figures contain no place marks on the axis. What is ‘T’ in the figures is not mentioned too. Type of errors presented in the figures is not specified. It would be much easy for reader if the values for each parameter on the figures would be marked with letters or asterisks to show significance of differences instead of using of bulky tables. Fillings of the columns in the histograms are almost indistinguishable.
13) Line 149: What is ‘accumulation of Pod under saline conditions’? Is it POD (peroxidase)? If so, how it can be accumulated? Did authors measured POD protein content or just activity?
14) Line 167-168: ‘In present study a significant increase was recorded in the activity of protease under salinity stress.’ I see no such an increase (Figure 1e). On the contrary, there is a tendency to decrease, and only after the treatment with 200 mg/L α-tocopherol it hardly significantly increased in Noori okra cultivar.
15) Lines 172-172: ‘Data revealed that total phenolics content significantly increased under salt stress in both okra varieties.’ Again, phenolics accumulated in Noori cultivar only after α-tocopherol fertigation (Figure 1f).
16) Line 177: ‘Significant accumulation of ascorbic acid content was recorded under salinity stress.’ Ascorbic acid content was very much the same as control in Sabzpari cultivar under salinity and decreased in Noori cultivar (Figure 1g). There was accumulation of ascorbate after α-tocopherol treatment, more significant in Noori cultivar.
17) Lines 205-206: ‘Malondialdehyde (MDA) content markedly enhanced under salt stress in both okra varieties.’ MDA content increased only in Sabzpari cultivar upon salinity (Figure 2d).
18) Lines 225-226: ‘Noori had higher fruit Ca2+ content than Sabzpari under both saline and salinity free regime.’ No, according to the data (Figure 2g), Noori had higher fruit Ca2+ content than Sabzpari only in control.
19) Line 232: What is ‘capsules plant 1’? Maybe, capsules plant -1?
20) The discussion is insufficient and inconsistent, authors mostly applied to literature citation than to consider their own data.
21) Line 265: Authors did not show ‘enhanced proline and GB content in fruit tissues of okra under salt stress in the current study’.
22) Line 285: Chloride level was not studied in the current study.
Taking all mentioned above into account, the manuscript would excellently merit publishing in Plants, but it requires considerable rewriting with thorough rearrangement of data presentation and comprehensive discussion. In present form the manuscript has to be rejected.
Author Response
Dear editor,
Greetings,
Thank you very much for your time and comments regarding our manuscript (plants-1237560). Our manuscript “Foliar Spray of Alpha-Tocopherol Modulates Antioxidant Potential of Okra Fruit under Salt Stress”
” has been revised carefully and here we are giving our response to the reviewers’ comments.We have improved the manuscript according to the reviewers’ comments and suggestions. All the revisions can be easily identified from manuscript in track changes.
Once again thanks for your co-operation and valuable comments and suggestion. Moreover, the effort of the reviewer is highly appreciated. We are hoping for pleasant response and further good comments (if any) from your side.
Yours truly,
Dr. Athar Mahmood
Department of Agronomy,
The University of Agriculture Faisalabad, Pakistan
*********************************************************************
We are thankful to editor and reviewers for timely completion of review process and providing us with valuable feedback.
Response to reviewer # 1:
Dear reviewer, we are grateful to you for your comments and suggestions for the improvement of the article. We are thankful to your efforts and time for highlight key points to further strengthen the idea.
Comments and Suggestions for Authors
The manuscript of paper by Naqve M. et al. “Foliar spray of alpha-tocopherol modulates antioxidant potential of okra fruit under salt stress” is devoted to the study of fertigation of vitamin E on the Abelmoschus esculentus plant to cope with salinity effects, mainly oxidative stress effects. Authors provided set of data obtained on antioxidant enzymes activities, antioxidant and osmolyte content as well as ion homeostasis upon α-tocopherol treatment under saline stress. Unfortunately, data are presented in unclear form and rather poorly discussed that in many cases it is very difficult to agree with author’s interpretation (see below). The manuscript contains lots of misformattings and design of figures, in spite of rather good English spelling.
1) There are several inaccuracies in the abstract. It is written ‘Current study was conducted twice to determine…’ (line 19). What does it mean? In ‘Material and Methods’ section it is said that there were 4 replicates, neither two. Different parameters were mentioned to alter ‘in both okra varieties’ (line 24). It is better to specify, what varieties? Abstract contains several missing spaces (lines 26, 31, 32).
Response: Twice means the experiment was conducted in two consecutive growing years
Twice does not mean two replicates, rather it is for two growing years and four replicates for each treatment. Required information is also incorporated (line 24).
2) Should not the ‘Material and Methods’ section be placed in the end part of the article before the ‘Conclusion’ section? Also there is a mistake in the word ‘methods’.
Response: ‘Material and Methods’ section has been placed in the end part of the article before the ‘Conclusion’ section as suggested. Word ‘methods’ has also been corrected.
3) Authors use several variants of ‘α-tocopherol’ mentions throughout the text (alpha tocopherol, alpha-tocopherol and α-Toc). It is better to unify the variant and use the abbreviation since the first mention, and not from the second page of the text.
Response: Abbreviation of alpha-tocopherol “α-Toc” is incorporated as suggested throughout the text to keep uniformity.
4) Line 44, does (6.5 dsm-1) mean (6.5 dSm-1)? If so, it has to be corrected.
Response: Yes, it is 6.5 dSm-1 and corrected in the text
5) There are several incorrect uses of uppercase letters in the text (lines 65, 263).
Response: Corrected
6) Information on lines 74-77 of ‘Introduction’ is somewhat similar to the information on lines 69-73 and has to be rearranged.
Response: Rearranged
7) It is unclear, how 100 mM concentration of NaCl) was maintained in aliquot parts of 50 mM (lines 94-96).
Response: The correction has been made in “Materials and Methods” section. In fact, just to avoid osmotic shocks to plants, the salinity level was maintained in two phases. In first phase (at start of salinity) 50 mM NaCl level was maintained and after two days of this level, the required level of 100 mM was maintained.
8) What is ‘Tween 20 @ 0.1%was’? There is missing space between ‘%’ and ‘was’ as well.
Response: For foliar application, to ensure maximum absorption we use surfactants. These are detergent like compounds which cause the breakdown of the cuticle (hydrophobic layer). Structure of guard cells only supports outward movement of water. The level 0.1 % is recommended level reported in many references.
9) Lines 104-105, plant material / buffer ratio and buffer pH are not specified.
Response: Plant material / buffer ratio and buffer pH are specified
10) Authors measured content of a number of antioxidants, but not α-tocopherol itself. Also glycine betaine can hardly be attributed to antioxidants.
Response: The protocol for the determination of α-tocopherol was not optimized in our lab. It is good suggestion and we will do it in next experiments.
11) Lines 126-128: Authors counted the number of capsules per plant in terms of yield assessment. Why not to measure capsules weight?
Response: We have missed the data for capsules weight
12) The results of the study are not clearly presented. There are 2 tables and 2 figures with excessive information given in absolutely reader-unfriendly way. It is better to split figures into several ones and enlarge their size. The figures contain no place marks on the axis. What is ‘T’ in the figures is not mentioned too. Type of errors presented in the figures is not specified. It would be much easy for reader if the values for each parameter on the figures would be marked with letters or asterisks to show significance of differences instead of using of bulky tables. Fillings of the columns in the histograms are almost indistinguishable.
Response: Lettering has been done and figures have been separated to more clear view for the readers as suggested by you. Type of error is actually standard deviation and also written in caption of figure. Split of Figures has been done as suggested by you. Other suggested corrections in figures are also done.
13) Line 149: What is ‘accumulation of Pod under saline conditions’? Is it POD (peroxidase)? If so, how it can be accumulated? Did authors measured POD protein content or just activity?
Response: Yes, it is POD (peroxidase) and corrected. Activity of POD was measured
14) Line 167-168: ‘In present study a significant increase was recorded in the activity of protease under salinity stress.’ I see no such an increase (Figure 1e). On the contrary, there is a tendency to decrease, and only after the treatment with 200 mg/L α-tocopherol it hardly significantly increased in Noori okra cultivar.
Response: Corrected as suggested
15) Lines 172-172: ‘Data revealed that total phenolics content significantly increased under salt stress in both okra varieties.’ Again, phenolics accumulated in Noori cultivar only after α-tocopherol fertigation (Figure 1f).
Response: Correction has been done
16) Line 177: ‘Significant accumulation of ascorbic acid content was recorded under salinity stress.’ Ascorbic acid content was very much the same as control in Sabzpari cultivar under salinity and decreased in Noori cultivar (Figure 1g). There was accumulation of ascorbate after α-tocopherol treatment, more significant in Noori cultivar.
Response: Corrected
17) Lines 205-206: ‘Malondialdehyde (MDA) content markedly enhanced under salt stress in both okra varieties.’ MDA content increased only in Sabzpari cultivar upon salinity (Figure 2d).
Response: Corrected
18) Lines 225-226: ‘Noori had higher fruit Ca2+ content than Sabzpari under both saline and salinity free regime.’ No, according to the data (Figure 2g), Noori had higher fruit Ca2+ content than Sabzpari only in control.
Response: Corrected
19) Line 232: What is ‘capsules plant 1’? Maybe, capsules plant -1?
Response: Corrected (capsules plant -1)
20) The discussion is insufficient and inconsistent, authors mostly applied to literature citation than to consider their own data.
Response: Discussion section has been improved
21) Line 265: Authors did not show ‘enhanced proline and GB content in fruit tissues of okra under salt stress in the current study’.
Response: Corrected
22) Line 285: Chloride level was not studied in the current study.
Response: We did not determine the chloride level. And it was erroneously mentioned in the discussion section along with Na+. Correction has been made in discussion section.

Reviewer 2 Report
Except for the English and minor typing errors the text is well written and really interesting. The results obtained are encouraging as a solution to cultivate crops in salt stress conditions and maybe just in stress conditions keeping a good yield.
In the discussion some references could be missing
In the attached document you will find detailed suggestions.
Maybe wider conclusions about the use of tocopherol to remediate the impact of salt stress to other crops could be expressed

Author Response
Dear editor,
Greetings,
Thank you very much for your time and comments regarding our manuscript (plants-1237560). Our manuscript “Foliar Spray of Alpha-Tocopherol Modulates Antioxidant Potential of Okra Fruit under Salt Stress”
” has been revised carefully and here we are giving our response to the reviewers’ comments.We have improved the manuscript according to the reviewers’ comments and suggestions. All the revisions can be easily identified from manuscript in track changes.
Once again thanks for your co-operation and valuable comments and suggestion. Moreover, the effort of the reviewer is highly appreciated. We are hoping for pleasant response and further good comments (if any) from your side.
Yours truly,
Dr. Athar Mahmood
Department of Agronomy,
The University of Agriculture Faisalabad, Pakistan
*********************************************************************
We are thankful to editor and reviewers for timely completion of review process and providing us with valuable feedback.
Response to reviewer # 2:
Dear reviewer, we are grateful to you for your comments and suggestions for the improvement of the article. We are thankful to your efforts and time for highlight key points to further strengthen the idea.
Except for the English and minor typing errors the text is well written and really interesting. The results obtained are encouraging as a solution to cultivate crops in salt stress conditions and maybe just in stress conditions keeping a good yield.
Comment: In the discussion some references could be missing
Response: References have been added and cross matched.
In the attached document you will find detailed suggestions.
Maybe wider conclusions about the use of tocopherol to remediate the impact of salt stress to other crops could be expressed
Response: Needful has been done

Reviewer 3 Report
The manuscript entitled "Foliar Spray of Alpha-Tocopherol Modulates Antioxidant Potential of Okra Fruit under Salt Stress" provides data regarding the effect of alpha-tocopherol as a foliar spray to improve the yield and biochemical constituents of fresh green capsules of okra grown under salt stress. Below you can see the minor modifications:
Line 18: As an antioxidant, alpha-tocopherol protects the plants from a salinity-induced oxidative burst.
Line 22: and yield, whereas increased H2O2 [...] and protease in both okra varieties was noticed.
Line 45: It remarkably affects the production of crops
Line 51: Salinity affects plant growth and yield by reducing photosynthesis rate
Line 76: Abbreviation should be firstly mentioned in the abstract. From there on using the abbreviation in the manuscript.
Line 78: As chloroplasts are sensitive to salinity stress [17], to combat salinity induced ROS, α-Toc works...
Line 251: did not increase its activity significantly
Line 268: enhanced the proline and GB content suggesting its role
Author Response
Dear editor,
Greetings,
Thank you very much for your time and comments regarding our manuscript (plants-1237560). Our manuscript “Foliar Spray of Alpha-Tocopherol Modulates Antioxidant Potential of Okra Fruit under Salt Stress”
” has been revised carefully and here we are giving our response to the reviewers’ comments.We have improved the manuscript according to the reviewers’ comments and suggestions. All the revisions can be easily identified from manuscript in track changes.
Once again thanks for your co-operation and valuable comments and suggestion. Moreover, the effort of the reviewer is highly appreciated. We are hoping for pleasant response and further good comments (if any) from your side.
Yours truly,
Dr. Athar Mahmood
Department of Agronomy,
The University of Agriculture Faisalabad, Pakistan
*********************************************************************
We are thankful to editor and reviewers for timely completion of review process and providing us with valuable feedback.
Response to reviewer # 3:
Dear reviewer, we are grateful to you for your comments and suggestions for the improvement of the article. We are thankful to your efforts and time for highlight key points to further strengthen the idea.
Comments and Suggestions for Authors
The manuscript entitled "Foliar Spray of Alpha-Tocopherol Modulates Antioxidant Potential of Okra Fruit under Salt Stress" provides data regarding the effect of alpha-tocopherol as a foliar spray to improve the yield and biochemical constituents of fresh green capsules of okra grown under salt stress. Below you can see the minor modifications:
Line 18: As an antioxidant, alpha-tocopherol protects the plants from a salinity-induced oxidative burst.
Response: Suggestions have been incorporated
Line 22: and yield, whereas increased H2O2 [...] and protease in both okra varieties was noticed.
Response: Corrected as you suggetsed
Line 45: It remarkably affects the production of crops
Response: Corretion done
Line 51: Salinity affects plant growth and yield by reducing photosynthesis rate
Response: Suggestions have been incorporated
Line 76: Abbreviation should be firstly mentioned in the abstract. From there on using the abbreviation in the manuscript.
Response: Change has been incorporated,
Line 78: As chloroplasts are sensitive to salinity stress [17], to combat salinity induced ROS, α-Toc works...
Response: done as you suggested
Line 251: did not increase its activity significantly
Response: correction is carried out as you suggested
Line 268: enhanced the proline and GB content suggesting its role
Response: Correction incorporated

Reviewer 4 Report
In presented manuscript, Naqve et al. investigated the influence of exogenous application of α-tocopherol on the antioxidant potential of okra plants. This work is interesting however in the present version, it can not be accepted for publication.
Introduction is poorly written and contains laconic statements. The methods are cursory described - in fact, for instance it is not known how many repetitions there were in each experiment. Discussion section is a description of the results and Conclusions should be also improved. Also, English should be definitely revised throughout the manuscript. Some sentences seem incomplete (e.g. Introduction, line 78). Considering the above, I decided to reject this manuscript in this version.
Author Response
Dear editor,
Greetings,
Thank you very much for your time and comments regarding our manuscript (plants-1237560). Our manuscript “Foliar Spray of Alpha-Tocopherol Modulates Antioxidant Potential of Okra Fruit under Salt Stress”
” has been revised carefully and here we are giving our response to the reviewers’ comments.We have improved the manuscript according to the reviewers’ comments and suggestions. All the revisions can be easily identified from manuscript in track changes.
Once again thanks for your co-operation and valuable comments and suggestion. Moreover, the effort of the reviewer is highly appreciated. We are hoping for pleasant response and further good comments (if any) from your side.
Yours truly,
Dr. Athar Mahmood
Department of Agronomy,
The University of Agriculture Faisalabad, Pakistan
*********************************************************************
We are thankful to editor and reviewers for timely completion of review process and providing us with valuable feedback.
Response to reviewer # 4:
Dear reviewer, we are grateful to you for your comments and suggestions for the improvement of the article. We are thankful to your efforts and time
Introduction is poorly written and contains laconic statements. The methods are cursory described - in fact, for instance it is not known how many repetitions there were in each experiment. Discussion section is a description of the results and Conclusions should be also improved. Also, English should be definitely revised throughout the manuscript. Some sentences seem incomplete (e.g. Introduction, line 78). Considering the above, I decided to reject this manuscript in this version.
Response:
Dear Reviewer,
Thank you very much for valuable comments/suggestions for the improvement of article. We have greatly revised the article as per reviewers comments and suggestions.
We hope article in resent form can be considered for publication.

Round 2
Reviewer 1 Report
Authors have very much improved the manuscript. They’ve taken into account majority of reviewer’s notes: improved methods, rearranged figures, reorganized discussion. Now the manuscript is almost ready to be accepted. But still it contains lots of misformattings: excessive dots (lines 119, 148) or dash (line 312), missing spaces (lines 18, 32, 35, 74, 178, 243, 353) or reference (line 286). Protease is not an antioxidant enzyme (line 26). After correction of mentioned remarks the paper could be recommended for publishing.
Author Response
Dear editor,
Greetings,
Thank you very much for your time and comments regarding our manuscript (plants-1237560). Our manuscript “Foliar Spray of Alpha-Tocopherol Modulates Antioxidant Potential of Okra Fruit under Salt Stress”
” has been revised carefully and here we are giving our response to the reviewers’ comments.We have improved the manuscript according to the reviewers’ comments and suggestions. All the revisions can be easily identified from manuscript in track changes.
Once again thanks for your co-operation and valuable comments and suggestion. Moreover, the effort of the reviewer is highly appreciated. We are hoping for pleasant response and further good comments (if any) from your side.
Yours truly,
Dr. Athar Mahmood
Department of Agronomy,
The University of Agriculture Faisalabad, Pakistan
*********************************************************************
We are thankful to editor and reviewers for timely completion of review process and providing us with valuable feedback.
Response to reviewer # 1:
Dear reviewer, we are grateful to you for your comments and suggestions for the improvement of the article. We are thankful to your efforts and time for highlight key points to further strengthen the idea.
COMMENT: Authors have very much improved the manuscript. They’ve taken into account majority of reviewer’s notes: improved methods, rearranged figures, reorganized discussion. Now the manuscript is almost ready to be accepted. But still it contains lots of misformattings: excessive dots (lines 119, 148) or dash (line 312), missing spaces (lines 18, 32, 35, 74, 178, 243, 353) or reference (line 286). Protease is not an antioxidant enzyme (line 26). After correction of mentioned remarks the paper could be recommended for publishing.
REPONSE: All proposed correction and some others have been corrected and mentioned in Blue font color (Track changes)

Reviewer 4 Report
Dear Authors,
the manuscript was only partially corrected. In the sections: Introduction, Discussion and Conclusions, the Authors did not respond to my comments - some minor errors were only corrected. In the Material and Methods chapter there are incomprehensible descriptions of the experiment, it is not known how many pots there were, what was the period of single experiment. What do 4 repetitions mean? 4 pots of 5 plants? Maybe it is included in ANOVA - hence I would like to see the results from the ANOVA table. Moreover, there are still many of text errors, e.g. missing spaces (only examples!: lines: 18, 32, 129, 136…), too many spaces (e.g. lines: 151, 158, 172), missing parentheses (e.g. lines 178, 182) etc. Hence, in my opinion, the manuscript has not been revised carefully. Taking the above - I cannot recommend this version of manuscript for publication.
Author Response
Dear editor,
Greetings,
Thank you very much for your time and comments regarding our manuscript (plants-1237560). Our manuscript “Foliar Spray of Alpha-Tocopherol Modulates Antioxidant Potential of Okra Fruit under Salt Stress”
” has been revised carefully and here we are giving our response to the reviewers’ comments.We have improved the manuscript according to the reviewers’ comments and suggestions. All the revisions can be easily identified from manuscript in track changes.
Once again thanks for your co-operation and valuable comments and suggestion. Moreover, the effort of the reviewer is highly appreciated. We are hoping for pleasant response and further good comments (if any) from your side.
Yours truly,
Dr. Athar Mahmood
Department of Agronomy,
The University of Agriculture Faisalabad, Pakistan
*********************************************************************
We are thankful to editor and reviewers for timely completion of review process and providing us with valuable feedback.
Response to reviewer # 4:
Dear reviewer, we are grateful to you for your comments and suggestions for the improvement of the article. We are thankful to your efforts and time for highlight key points to further strengthen the idea.
COMMENT 1: The manuscript was only partially corrected. In the sections: Introduction, Discussion and Conclusions, the Authors did not respond to my comments – some minor errors were only corrected.
RESPONSE: Dear Reviewer, you did not mention specific corrections you only asked to improve the above said section, we improved with needful corrections and amendments in sections you mentioned.
COMMENT 2: In the Material and Methods chapter there are incomprehensible descriptions of the experiment, it is not known how many pots there were, what was the period of single experiment. What do 4 repetitions mean? 4 pots of 5 plants? Maybe it is included in ANOVA - hence I would like to see the results from the ANOVA table.
RESPONSE: a). There were 64 pots for single experiment
b). The period of single experiment was of six months (we also added in methodology now)
c). We mentioned 4 replications (its mean the treatments were repeated 4 times in single experiment). There were 64 pots and each pot contain 5 plants)
(2 varieties x 2 salt levels x 4 Alpha-Tochopherol levels x 4 replicates = 64 pots)
- d) I hope now, if you look into ANOVA table you will understand these mentioned things
COMMENTS: Moreover, there are still many of text errors, e.g. missing spaces (only examples!: lines: 18, 32, 129, 136…), too many spaces (e.g. lines: 151, 158, 172), missing parentheses (e.g. lines 178, 182) etc. Hence, in my opinion, the manuscript has not been revised carefully.
REPONSE: All proposed correction and some others have been corrected and mentioned in Blue font color (Track changes)

Round 3
Reviewer 4 Report
I am satisfied with the revision of the manuscript. Hence, I can recommend the current version of the manuscript for publication.
Author Response
Thank you for your kind support for the article under consideration.